# Unexpected and Synergistical Effects of All-Trans Retinoic Acid and TGF-β2 on Biological Aspects of 2D and 3D Cultured ARPE19 Cells

**DOI:** 10.3390/biomedicines12102228

**Published:** 2024-09-30

**Authors:** Megumi Higashide, Megumi Watanabe, Tatsuya Sato, Toshifumi Ogawa, Araya Umetsu, Soma Suzuki, Masato Furuhashi, Hiroshi Ohguro, Nami Nishikiori

**Affiliations:** 1Departments of Ophthalmology, Sapporo Medical University, S1W17, Chuo-ku, Sapporo 060-8556, Japan; megumi.h@sapmed.ac.jp (M.H.); watanabe@sapmed.ac.jp (M.W.); araya.alaya.favreweissth@gmail.com (A.U.); ophthalsoma@sapmed.ac.jp (S.S.); ooguro@sapmed.ac.jp (H.O.); 2Departments of Cardiovascular, Renal and Metabolic Medicine, Sapporo Medical University, S1W17, Chuo-ku, Sapporo 060-8556, Japan; sato.tatsuya@sapmed.ac.jp (T.S.); a08m024@yahoo.co.jp (T.O.); furuhasi@sapmed.ac.jp (M.F.); 3Departments of Cellular Physiology and Signal Transduction, Sapporo Medical University, S1W17, Chuo-ku, Sapporo 060-8556, Japan

**Keywords:** TGF-β2, human retinal pigment epithelium, 3D culture, ATRA, hypoxia

## Abstract

**Objectives:** To study the effects of all-trans retinoic acid (ATRA) on TGF-β2-induced effects of human retinal pigment epithelium cells under normoxia and hypoxia conditions. **Methods:** Two-dimensionally (2D) and three-dimensionally (3D) cultured ARPE19 cells were subjected to cellular functional analyses by transepithelial electrical resistance (TEER) and an extracellular flux assay (2D), measurement of levels of reactive oxygen species (ROS), gene expression analyses of *COL1*, *αSMA*, *Zo-1*, *HIF1α*, and *PGC1α* (2D), and physical property analyses (3D). **Results:** Under a normoxia condition, treatment with 100 nM ATRA substantially decreased barrier function regardless of the presence of 5 ng/mL TGF-β2 in 2D ARPE19 monolayer cells. Under a hypoxia condition, treatment with ATRA conversely increased barrier function, but the effect was masked by a marked increase in effects induced by TGF-β2. Although ATRA alone did not affect cellular metabolism and ROS levels in 2D ARPE cells, treatment with ATRA under a hypoxia condition did not affect ROS levels but shifted cellular metabolism from mitochondrial respiration to glycolysis. The changes of cellular metabolism and ROS levels were more pronounced with treatment of both ATRA and TGF-β2 independently of oxygen conditions. Changes in mRNA expressions of some of the above genes suggested the involvement of synergistical regulation of cellular functions by TGF-β2 and hypoxia. In 3D ARPE spheroids, the size was decreased and the stiffness was increased by either treatment with TGF-β2 or ATRA, but these changes were unexpectedly modulated by both ATRA and TGF-β2 treatment regardless of oxygen conditions. **Conclusions:** The findings reported herein indicate that TGF-β2 and hypoxia synergistically and differentially induce effects in 2D and 3D cultured ARPE19 cells and that their cellular properties are significantly altered by the presence of ATRA.

## 1. Introduction

Human retinal pigment epithelial (RPE) cells are not only physiologically required for maintenance of homeostasis of posterior segments of the eye, such as the retina and choroid, but also function as a putative outer blood retinal barrier (oBRB) with the ocular choroid [1]. Simultaneously, RPE contributes to ocular pathogenesis such as retinal and choroidal neovascularization and fibrosis, resulting in proliferative vitreoretinopathy (PVR) and age-related macular degeneration (ARMD) [2]. In terms of the causative etiology of these diseases, transforming growth factor-beta 2 (TGF-β2)-induced effects of RPE cells have been identified as a key supporting molecular pathogenesis [3,4,5,6]. In addition to TGF-β2, hypoxia-related mechanisms are also known as important factors for the progression of ocular diseases, including ARMD, diabetic retinopathy (DR), ischemic type of retinal-vein occlusion (RVO), and retinopathy of prematurity (ROP) [7,8,9], by facilitating the secretion of various antigenic cytokines such as vascular endothelial growth factor (VEGF). Therefore, both TGF-β2-induced effects of RPE cells and hypoxia are important for understanding the etiology of these diseases and, in turn, providing valuable suggestions for the development of new therapeutic strategies for these diseases.

All-trans-retinoic acid (ATRA), a derivative of vitamin A, is known to potently regulate the growth and differentiation of various sources of cells [10,11]. Biologically, ATRA exerts anti-inflammatory and antifibrotic activities by inhibiting nuclear factor–κB (NF-κB) [12] and TGF-β [13]-related signaling, respectively. For instance, it was revealed that ATRA inhibited TGF-β–induced liver fibrosis by down-regulating the collagen 1A2 gene [14] and that 9-Cis-retinoic acid, an isomer of ATRA, attenuated TGF-β–induced fibrotic changes of human mesangial cells [13]. In the field of ophthalmology, in vitro studies have suggested that anti-TGF-β effects of ATRA are promising candidates for regulating postoperative TGF-β-induced subconjunctival fibrosis [15,16]. However, a recent study showed that ATRA stimulates secretion of TGF-β2 by RPE cells via the phospholipase C signaling pathway but not by the adenylyl cyclase signaling pathway [17]. Another study showed that ATRA inhibits the expression of RDH5 and enhances the expression of TGF-β2, suggesting possible involvement of RDH5 in TGF-β2-related effects of RPE cells mediated by ATRA [18]. In addition, it was shown that ATRA had protective effects against hypoxia-induced changes in malignant tumors [19]. Collectively, since ATRA is controversially affected by TGF-β2-related effects of conjunctiva and RPE and its effects may be influenced by hypoxia, an additional study will be required to elucidate ATRA-induced effects on pathogenic conditions of RPE cells induced by TGF-β2 and/or hypoxia using a reliable in vitro model to mimic spatially spreading proliferative changes of RPE cells. For this purpose, in our preceding study we developed an in vitro three-dimensional (3D) cultured model [20,21,22,23] in addition to the conventional two-dimensional (2D) cultured model.

In the current study, using these models, effects of ATRA on TGF-β2-induced effects of ARPE19 cells under normoxia and hypoxia conditions were determined by the following analyses: (1) barrier functions of 2D ARPE19 monolayers (2D), (2) cellular mitochondrial and glycolytic functions (2D), (3) analysis of ROS levels, (4) expression of the major ECM *protein collegen1 (COL1)*, *α smooth muscle actin (αSMA)*, *hypoxia-related factors α*; *HIF 1α* and *PGC1a*, and the tight-junction related molecule *Zo1*(2D), and (5) physical properties of 3D spheroids.

## 2. Materials and Methods

### 2.1. 2D and 3D Cultures of Human Retinal Pigment Epithelium Cells, ARPE19

Based on the compliance with the tenets of the Declaration of Helsinki, all experimental protocols using a human-derived cell, a commercially available human retinal pigment epithelium cell line, ARPE19 (#CRL-2302™, ATCC, Manassas, VA, USA), were performed after approval by the internal review board of Sapporo Medical University. In brief, as described previously [24], ARPE19 cells were 2D cultured in a 150 mm planar culture dish until 90% confluence at 37 °C in an HG-DMEM (Wako, Osaka, Japan) containing 10% FBS (Biosera, Cholet, France), 1% L-glutamine (Wako, Osaka, Japan), and 1% antibiotic-antimycotic (Thermo Fisher Scientific, Tokyo, Japan), and the cells were maintained by changing the medium every other day under a standard normoxia condition (37 °C, 5% CO_2_, 20% O_2_) or hypoxia condition (37 °C, 5% CO_2_, 1% O_2_). Those 2D cultured cells were further processed for a 3D spheroid culture in a hanging drop culture plate (# HDP1385, Sigma-Aldrich, St. Louis, MO, USA) (Day 0) using the above 2D culture medium supplemented with 0.25% methylcellulose (Methocel A4M, Sigma-Aldrich, St. Louis, MO, USA). On each following day, half of the spheroid medium was replaced by a fresh medium, and the cultures were maintained until Day 6.

At Day one, 5 ng/mL TGF-β2 (Wako, Osaka, Japan) was added to the growth medium or spheroid medium to stimulate EFFECTS of 2D or 3D cultured cells, respectively, in the absence or presence of 100 nM ATRA (Wako, Osaka, Japan). The concentrations of TGF-β2 and ATRA were followed by previous studies [24,25].

### 2.2. Barrier Function of 2D ARPE19 Monolayer by TEER

To estimate the barrier function of a 2D ARPE19 monolayer cultured in a TEER plate (0.4 μm pore in size and 12 mm in diameter; Corning Transwell, Sigma-Aldrich, St. Louis, MO, USA), measurement of the TEER value was performed by using an electrical resistance measurement system (KANTO CHEMICAL CO. INC., Tokyo, Japan) as described previously [23,26].

### 2.3. Measurement of Real-Time Cellular Metabolic Functions

The oxygen consumption rate (OCR) and the extracellular acidification rate (ECAR) in 2D monolayer ARPE19 cells that were not treated with (Control) or were treated with 5 ng/mL TGF-β2 and/or 100 nM ATRA under a normoxia condition (~21% O_2_) and a hypoxia (~3% O_2_) condition were determined by using a Seahorse XFe96 Bioanalyzer (Agilent Technologies, Santa Clara, CA, USA) according to the manufacturer’s instructions. In brief, 20,000 2D ARPE cells that were treated with ATRA and/or TGF-β2 for five days as described above were placed in each well of an XFe96 Cell Culture Microplate (#103794-100, Agilent Technologies, Santa Clara, CA, USA) on the day before the assay. On the day of the assay, the culture medium was replaced with 180 μL of Seahorse XF DMEM assay medium (#103575-100, pH 7.4, Agilent Technologies, Santa Clara, CA, USA) supplemented with 5.5 mM glucose, 2.0 mM glutamine, and 1.0 mM sodium pyruvate. The plates were then incubated in a CO_2_-free incubator at 37 °C for 30 min prior to the assay under a normoxia or hypoxia condition, depending on the assay conditions. OCR and ECAR were determined by using a Seahorse XFe96 Bioanalyzer at baseline and after sequential injection of oligomycin (final concentration: 2.0 μM), carbonyl cyanide p-trifluoromethoxyphenylhydrazone (FCCP, final concentration: 5.0 μM), and rotenone/antimycin. A mixture (final concentration: 1.0 μM). The values were normalized to the amount of protein per well assessed by a BCA protein assay (TaKaRa Bio, Siga, Japan).

For metabolic indices, the following formulas were used. Baseline OCR was determined by subtracting OCR with rotenone/antimycin A from OCR at baseline; ATP-linked respiration was determined by subtracting OCR with oligomycin from OCR at baseline; proton leak was determined by subtracting OCR with rotenone/antimycin A from OCR with oligomycin; maximal respiration was determined by subtracting OCR with rotenone/antimycin A from OCR with FCCP; non-mitochondrial respiration was determined by OCR with rotenone/antimycin A; and baseline OCR/ECAR was determined by dividing the average value of OCR at baseline by the average value of ECAR at baseline.

### 2.4. Measurement of Reactive Oxygen Species (ROS) Levels

To calculate 2D cultured ARPE19 cell function in each treatment, reactive oxygen species (ROS) levels were determined using a ROS assay kit (DOJINDO, Kumamoto, Japan). Briefly, after discarding the culture medium, cells were washed twice with Hanks’ Balanced Salt Solution (HBSS, Thermo Fisher Scientific, Waltham, MA, USA), and 100 μL of Highly Sensitive DCFH-DA Working Solution was added to the cells. The cells were incubated for 30 min in an incubator set at 37 °C, equilibrated with 20% or 1% O_2_ conditions. The working solution was then discarded, and the cells were washed twice with HBSS. HBSS was added to the wells, and fluorescence signals were observed using a fluorescence plate reader (Excitation: 490–520 nm, Emission: 510–540 nm).

### 2.5. Measurements of Physical Property, Size, and Solidity of 3D ARPE19 Spheroids

As the physical properties, size, and solidity of the 3D ARPE19 spheroids, their largest cross-sectional area (CSA) of a phase contrast image of a 3D ARPE19 spheroid obtained by an inverted microscope (Nikon ECLIPSE TS2, Tokyo, Japan) and the requiring force (μN) to compress a living 3D ARPE19 spheroid during 20 s using a micro-squeezer (MicroSquisher, CellScale, Waterloo, ON, Canada), respectively, were measured as reported previously [20,22].

### 2.6. Other Analytical Methods

Total RNA was extracted from the 2D or 3D cultured ARPE19 cells, and reverse transcription and real-time PCR were carried out as previously reported [21,27] using specific primers and probes (Appendix A). All data are shown as arithmetic means ± the standard error of the mean (SEM). Differences between groups were evaluated by one-way or two-way analysis of variance (ANOVA). When ANOVA indicated a significant overall difference, multiple comparisons of the groups were further performed by the Tukey’s Honest Significant Difference (HSD) post-hoc test. All statistical analyses were performed using GraphPad Prism version 8 or 9 (GraphPad Software, San Diego, CA, USA), depending on the experiments as described in recent reports [21,27].

## 3. Results

### 3.1. Effects of ATRA on the Barrier Function of TGF-β2-Teated or UNTREATED ARPE19 Cell Monolayer under Different Oxygen Conditions

To evaluate the effects of ATRA on the barrier function as a putative outer blood retinal barrier (oBRB) of RPE cells, 2D ARPE19 cell monolayers were subjected to a TEER measurement (Figure 1). As demonstrated in our precedent study, TGF-β2 treatment caused no change in the TEER values under a normoxia condition and a significant increase in the TEER values under a hypoxia condition. In a 2D ARPE19 monolayer not treated with TGF-β2, TEER values were significantly decreased or increased by ATRA under a normoxia condition or a hypoxia condition, respectively. However, the TEER values of a TGF-β2-treated 2D ARPE19 monolayer were substantially decreased under both normoxia and hypoxia conditions. Collectively, these results suggested that ATRA-induced effects on a 2D ARPE19 monolayer were markedly modulated by treatment of TGF-β2 and/or different oxygen conditions.

### 3.2. Effects of ATRA on Cellular Metabolic Functions and ROS Levels of TGF-β2-Teated or Untreated 2D Cultured ARPE-19 Cells under Different Oxygen Conditions

Next, to study the metabolic effects (Figure 2) and ROS levels (Figure 3) of ATRA on the TGF-β2-induced effects of RPE cells under different oxygen conditions, 2D cultured ARPE19 cells treated or not treated with TGF-β2 were prepared in the absence or presence of 100 nM ATRA under normoxia and hypoxia conditions and were subjected to an extracellular flux assay as shown in Figure 2. Under a normoxia condition, the effect of TGF-β2 or ATRA on cellular metabolism was minor. Treatment with TGF-β2 reduced non-mitochondrial respiration and induced a shift from mitochondrial respiration to glycolysis, while such an effect was not observed with treatment with ATRA alone. In contrast, treatment with both ATRA and TGF-β2 reduced most of the mitochondria-related metabolic indices and induced a marked shift from mitochondrial respiration to glycolysis under a normoxia condition. Interestingly, under a hypoxia condition, ATP-linked respiration and maximal respiration, which are major mitochondrial functions, were predominantly reduced by even ATRA treatment alone. In contrast, mitochondrial respiration was significantly shifted toward glycolysis by co-treatment with ATRA and TGF-β2, while TGF-β2 alone did not induce such metabolic changes under a hypoxia condition.

In terms of ROS levels, although those of nontreated ARPE19 cells were relatively increased or decreased by hypoxia conditions or TGF-β2 alone, those were significantly decreased by both TGF-β2 and ATRA (Figure 3). These results suggested that metabolic changes and ROS levels induced by TGF-β2 or hypoxia are significantly and synergistically altered by treatment with ATRA.

To elucidate this issue further, mRNA expression levels of possibly related factors, including (A) a major ECM, *collagen 1* (*COL-1*), (B) a marker of myofibroblast formation [28], *α smooth muscle actin* (*αSMA*), (C) a major tight junction-related component, *Zo-1*, (D) *hypoxia-induced factor 1α* (*HIF1α*), and (E) a master regulator for mitochondrial respiration, peroxisome *proliferator-activated receptor gamma coactivator 1α* (*PGC1α*), were evaluated (Figure 4). As observed in our precedent study [24], 5 ng/mL TGF-α2 induced (1) significant up-regulation of *COL1* under both normoxia and hypoxia conditions and (2) significant down-regulation of *PGC1α* under a normoxia condition. Alternatively, 100 nM ATRA induced only up-regulation of *αSMA* under a hypoxia condition but unexpectedly induced significant increases in *HIF1α* and *Zo1* expression of TGF-β2-treated ARPE19 cells under both normoxia and hypoxia conditions. Based on these qPCR data together with functional data of TEER, seahorse cellular metabolic measurements, and ROS levels, we speculated the presence of some synergistical effects of ATRA and TGF-β2 in 2D cultured ARPE19 cells.

### 3.3. Effects of ATRA on Physical Properties of TGF-β2-Teated or Untreated 3D ARPE19 Spheroids under Different Oxygen Conditions

To further study the effects of ATRA on the spatially spreading TGF-β2-related effects of RPE cells, physical properties, including size and stiffness, of 3D ARPE19 spheroids were compared among the above conditions. As shown in Figure 5 and Figure 6, the sizes of 3D ARPE spheroids were significantly decreased by monotreatment of 5 ng/mL of TGF-β2 or 100 nM ATRA under a normoxia condition, and those effects were more pronounced under a hypoxia condition. However, unexpectedly, both 100 nM ATRA and 5 ng/mL TGF-β2 synergistically induced relative enlargement of 3D ARPE19 spheroids under normoxia and hypoxia conditions.

Such unexpected results indicating synergistic effects were also obtained for the stiffness of 3D ARPE19 spheroids (Figure 7), that is, (1) 100 nM ATRA or 5 ng/mL TGF-β2 caused a significant increase in the stiffness of 3D ARPE19 spheroids under both normoxia and hypoxia conditions, but (2) 100 nM ATRA had no effect or significant inhibitory effect under a normoxia condition and a hypoxia condition, respectively, on the TGF-β2-induced increase in stiffness.

**Figure 7 biomedicines-12-02228-f007:**
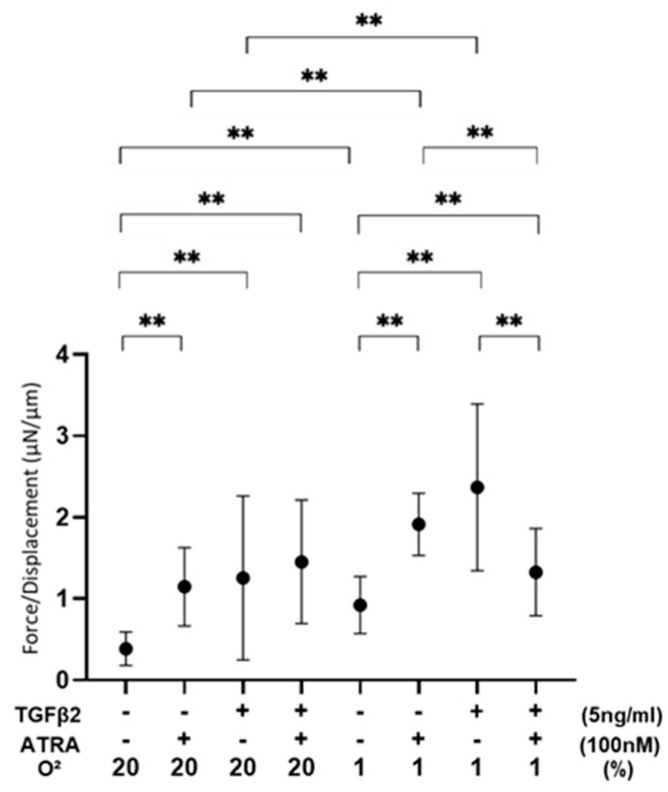
Effects of ATRA on 3D spheroid stiffness of 3D ARPE19 spheroids not treated or treated with TGF-β2 under normoxia and hypoxia conditions. ** *p* < 0.01. In the absence or presence of ATRA (100 nM), 3D ARPE19 spheroids not treated or treated with TGF-β2 (5 ng/mL) were prepared under normoxia and hypoxia conditions. The physical solidity of the 3D ARPE19 spheroids was analyzed by a micro-squeezer and the force required to produce a 50% deformity of a single spheroid during a period of 20 s (μN/μm) was plotted. All experiments were performed in triplicate using fresh preparations consisting of 16 spheroids each. ** *p* < 0.01.

**Figure 8 biomedicines-12-02228-f008:**
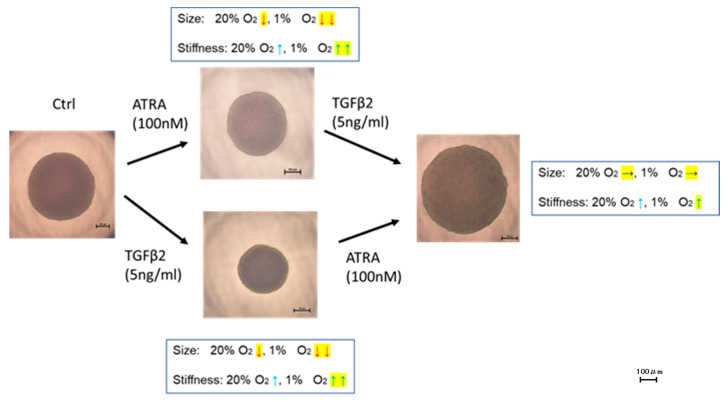
Unexpected effects of ATRA and TGF-β2 on physical properties of 3D ARPE19 spheroids. Unexpected effects of ATRA and TGF-β2 on the physical properties of ARPE19 spheroids are summarized. ↓: decrease (*p* > 0.05), ↓↓: decrease (*p* > 0.01), ↑: increase (*p* > 0.05), ↑↑: increase (*p* > 0.01), →: no significant change. Unexpected effects are shown in yellow-highlighted symbols. Scale bar: 100 μm.

## 4. Discussion

Retinoic acid (RA), a bioactive enzymatic metabolite of vitamin A, is involved in multifaceted regulations of cell growth and differentiation in various tissues and is essential for morphogenesis during the development of the eyes [29,30]. It was shown that vitamin A deficiency induces severe congenital eye defects [31,32,33,34] and that functional units of a heterodimer of retinoic acid receptors (RAR) and retinoid X receptors (RXR) are expressed in the developing eye [35,36]. In addition to the role of RA in normal development of the eyes, RA functions to promote the formation of tight junctions and thereby regulates the development and maintenance of the blood-brain barrier (BBB) [37] as well as the oBRB [38]. In fact, our group previously reported that ATRA caused significant reductions of vascular leakage in the diabetic retina by reinforcement of tight junction integrity of RPE [39]. ATRA-related beneficial effects have also been reported. For instance, ATRA inhibited the proliferation of human RPE cells in surgically excised PVR membranes [40], and intravitreal administration of ATRA prevented experimental PVR formation in animal models [41,42]. As possible underlying mechanisms of these beneficial effects of ATRA, it was suggested that ATRA suppresses migration and invasion abilities of RPE cells [43] by ECM remodeling effects [44]. However, in contrast, ATRA increases oxidative stress to induce cell death in ARPE19 cells [45], suggesting that ATRA may have both beneficial and unfavorable effects on ARPE19 cells, and those effects may be caused by various factors such as TGF-β2 and/or hypoxia stimulations. Since we found in our preceding study that TGF-β2-induced effects of both 2D and 3D cultured ARPE19 cells were significantly modified by hypoxia, those effects of ARPE19 cells may be altered by local environments such as normoxia or hypoxia and by planar or spatial spreading [24] under in vivo conditions. Therefore, it was of great interest to investigate the effects of ATRA on ARPE19 cells under such diverse conditions. In the current study, we observed some unexpected synergistic effects of ATRA on TGF-β2-induced effects in in vitro 3D cultured ARPE19 cell models (Figure 8). That is, (1) the 3D ARPE19 spheroids were significantly downsized by monotreatment of TGF-β2 or ATRA, whereas those sizes were not changed by both TGF-β2 and ATRA regardless of oxygen conditions, and (2) the stiffness of the TGF-β2 treated 3D ARPE19 spheroids was not altered by ATRA under a normoxia condition and was significantly decreased by ATRA under a hypoxia condition despite the fact that monotreatment of ATRA or TGF-β2 increased stiffness. Similar to those results, we found also synergistic and beneficial effects of ATRA on TGF-β2-treated 2D and 3D cultured HTM cells, which are in vitro models replicating a single sheet structure and a multiple sheet structure of the HTM [22,46,47,48,49].

Comparison of cellular metabolism between 2D cells and 3D spheroids is intriguing. Although underlying mechanisms inducing such different effects of ATRA and/or TGF-β2 of ARPE19 cells and HTM cells between 2D and 3D culture conditions remain to be elucidated and that direct comparison between 2D cells and 3D spheroids is challenging be-cause of the presence of numerous different experiment conditions using the extracellular flux analyzer system, our previous studies showed that 3D spheroid culture induced spontaneous adipogenic differentiation of 3T3-L1 preadipocytes and a significant metabolic shift from glycolysis to mitochondrial respiration in differentiated 3T3-L1 cells, but those were not observed in the 2D cultured condition, suggesting that 3D cultured conditions served preferable environments for their adipogenic differentiation [50]. In addition, in our previous study using ARPE19 cells, we also reported that cellularmetabolism in response to TGF-β2 was different between 2D cells and 3D spheroids [24]. In that study, PGC-1α, which is one of the master regulators of mitochondrial biogenesis, was significantly increased in 3D spheroids of ARPE19 compared to 2D cells of ARPE19, presumably partly via hypoxic stimulation. This suggests that mitochondrial OXPHOS may be more prominently activated in 3D spheroids under hypoxic conditions. However, it should be noted that the mechanisms of metabolic changes in 3D spheroids are different depending on the types of cells [50,51,52,53]. Furthermore, an RNA sequencing analysis suggested that STAT3 signaling may induce these specific phenotypes observed in 3D cultured 3T3-L1 cells [54]. In fact, previous studies have shown that STAT3 signaling mechanisms are crosslinked with TGF-β2 [55], hypoxia-related conditions [56], and ATRA [57,58]. Collectively, those synergistic effects of ATRA and TGF-β2 may support the well-accepted evidence that ATRA is involved in multifunctional and complicated effects in a variety of pathophysiological processes [11], despite the fact that metabolic adaptation in the 3D spheroids of ARPE19 cells has not been fully understood yet. Further study will be needed to elucidate the mechanisms why cellular metabolism in 3D spheroids shows diverse phenotypes.

HIF1, a nuclear factor bound to a cis-acting hypoxia response element (HRE), was found to be mainly involved in the induction of pathogenic conditions by hypoxia [59]. Such HIF-induced mechanisms are stimulated in the pathogenesis of epithelial mesenchymal transition (EMT) in RPE cell-related retinal diseases. For instance, knockdown of HIF1α in RPE cells inhibited the overexpression of VEGF and intercellular adhesion molecule 1 (ICAM-1), thereby substantially reducing vascular leakage and the CNV area in a laser CNV mouse model [60]. In addition, HIF1α-linked angiogenesis was also recognized as an important mechanism for the molecular pathogenesis of diabetic retinopathy [61,62,63]. Furthermore, it has been shown that HIF1α promotes TGF-β2-induced effects of human lens epithelial cells [64] and ARPE-19 cells [65]. In terms of the biological aspect of HIFs, activation of HIFs not only regulates mitochondrial respiration and oxidative stress but is also conversely regulated by mitochondrial metabolism, respiration, and oxidative stress [66]. In fact, it was reported that inhibition of PCG1α, a master regulator of mitochondrial function, caused deterioration of mitochondrial functions and stimulated an EMT response of ARPE19 cells [2]. It was also shown that TGF-β2-stimulated EMT of RPE cells induced significant down-regulation of PGC1α and mitochondrial dysfunctions, that is, a metabolic shift to reduced OXPHOS and increased glycolysis [67]. In the current study, mRNA expression of HIF1α was also synergistically and substantially up-regulated by TGF-β2 and ATRA, although TGF-β2 or ATRA alone had insignificant effects under both normoxia and hypoxia conditions, as similarly observed in the 3D spheroid stiffness analysis described above.

However, as of this writing, we do not know the reason for these unexpected synergistic effects of TGF-β2 and hypoxia, and therefore, to better understand these unidentified issues, further investigations using (1) an RNA-sequencing analysis, (2) specific agonists and antagonists against down-stream factors of related signaling, and (3) in vivo experiments using rodent models without or with degenerating retina will be required as our next project.

## 5. Conclusions

TGF-β2 and hypoxia synergistically and differentially induce effects in 2D and 3D cultured ARPE19 cells, and their cellular properties are significantly altered by the presence of ATRA. The complexity of these interactions highlights the need for further research to better understand the underlying mechanisms.

## Figures and Tables

**Figure 1 biomedicines-12-02228-f001:**
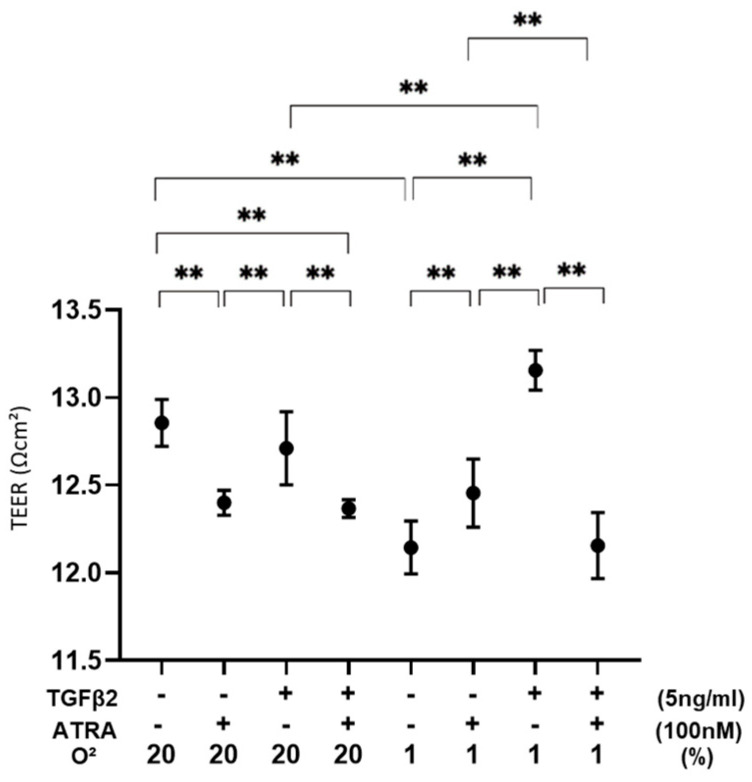
Effects of ATRA on trans-endothelial electrical resistance (TEER) values of 2D ARPE19 cell monolayers not treated or treated with TGF-β2 under normoxia and hypoxia conditions. ** *p* < 0.01. In the absence or presence of ATRA (100 nM), 2D ARPE19 cell monolayers not treated or treated with TGF-β2 (5 ng/mL) were prepared under normoxia and hypoxia conditions. The 2D cultures of ARPE19 cell monolayers at Day 6 were subjected to barrier function analyses by electric resistance (Ωcm^2^) measurements using TEER, and those values were plotted. All experiments were performed in triplicate using fresh preparations. ** *p* < 0.01.

**Figure 2 biomedicines-12-02228-f002:**
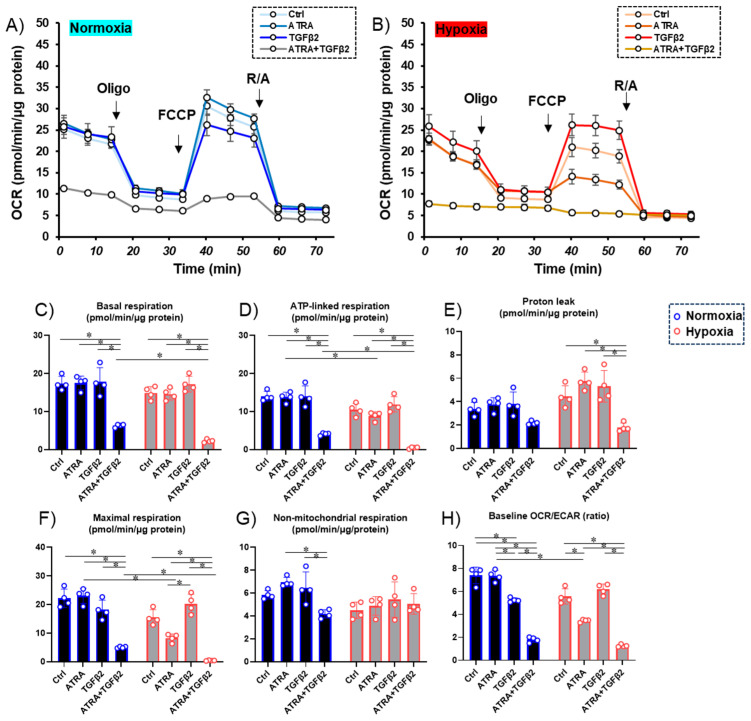
Effects of ATRA on cellular metabolic functions of 2D ARPE monolayer cells not treated or treated with TGF-β2 under normoxia and hypoxia conditions. * *p* < 0.05. Planar ARPE cells that were prepared with TGF-β2 (5 ng/mL) and/or ATRA (100 nM) for six days under normoxia and hypoxia conditions were subjected to real-time metabolic function analysis using a Seahorse XFe96 Bioanalyzer. Oxygen consumption rate (OCR) under a normoxia condition (~21% O_2_) and a hypoxia condition (~3% O_2_) and values of metabolic indices were determined. Panel (**A**) Changes in OCR under normal oxygen conditions. Panel (**B**) Changes in OCR under hypoxic conditions; Panels (**C**–**H**) Indices of mitochondrial functions. To avoid multiple comparisons, statistical analysis was performed using a two-way ANOVA with Tukey’s HSD post-hoc test for each mitochondrial function index. Ctrl = Control, Oligo = oligomycin, R/A = rotenone/antimycin A. Fresh preparations were used in all experiments (n = 4 in each group). * *p* < 0.05.

**Figure 3 biomedicines-12-02228-f003:**
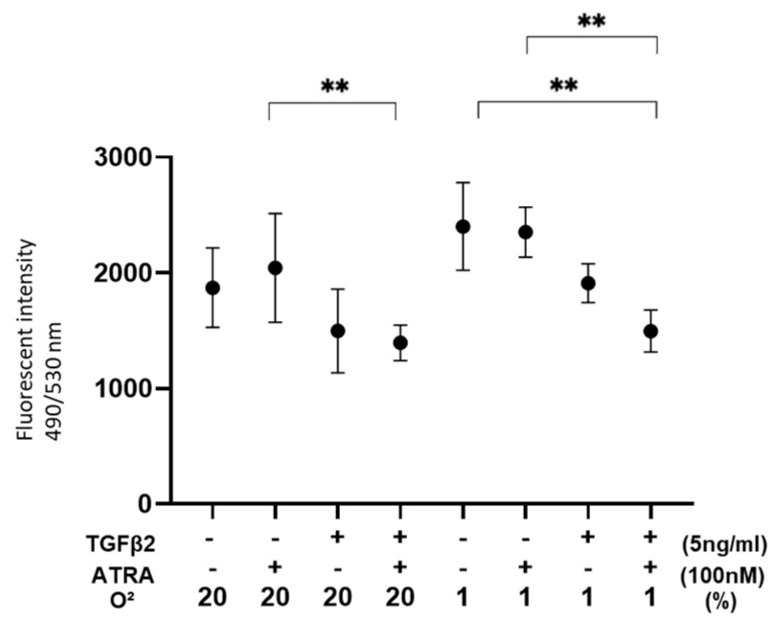
Effects of ATRA on ROS levels of 2D cultured ARPE19 cells not treated or treated with TGF-β2 under normoxia and hypoxia conditions. ** *p* < 0.01. In the absence or presence of ATRA (100 nM), 2D ARPE19 cells not treated or treated with TGF-β2 (5 ng/mL) were prepared under normoxia and hypoxia conditions. Each ARPE19 cell was then subjected to measurement of ROS, and those values were plotted. All experiments were performed in triplicate using fresh preparations. ** *p* < 0.01.

**Figure 4 biomedicines-12-02228-f004:**
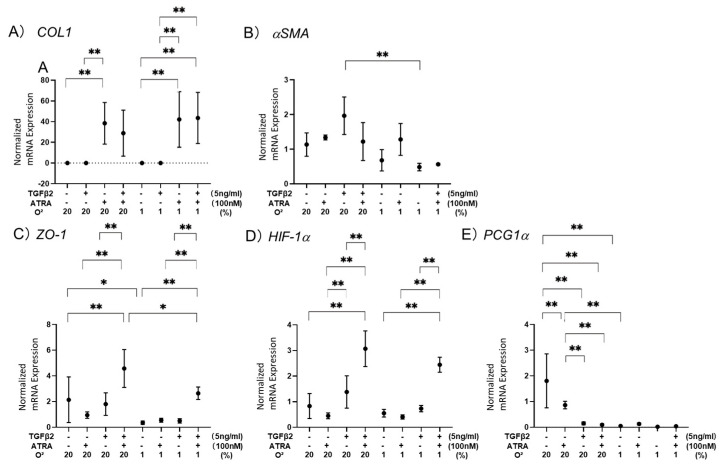
Effects of ATRA on mRNA expression of (**A**) *COL1*, (**B**) *αSMA*, (**C**) *Zo-1*, (**D**) *HIF1α*, and (**E**) *PGC1α* in 2D cultured ARPE19 cells not treated or treated with TGF-β2 under normoxia and hypoxia conditions. * *p* < 0.05, ** *p* < 0.01. In the absence or presence of ATRA (100 nM), 2D cultured ARPE19 cells not treated or treated with TGF-β2 (5 ng/mL) were prepared under normoxia and hypoxia conditions. Each sample was subjected to qPCR analysis, and the mRNA expression levels of possibly related factors, including Panel (**A**) a major ECM, *collagen 1* (*COL1*), Panel (**B**) a marker of myofibroblast formation, *α smooth muscle actin* (*αSMA*), Panel (**C**) a major tight junction-related component, *Zo-1*, Panel (**D**) *hypoxia-induced factor 1α* (*HIF1α*), and Panel (**E**) a master regulator for mitochondrial respiration, *peroxisome proliferator-activated receptor gamma coactivator 1α* (*PGC1α*), were estimated. All experiments were performed in duplicate using fresh preparations (n = 5 each). * *p* < 0.05, ** *p* < 0.01.

**Figure 5 biomedicines-12-02228-f005:**
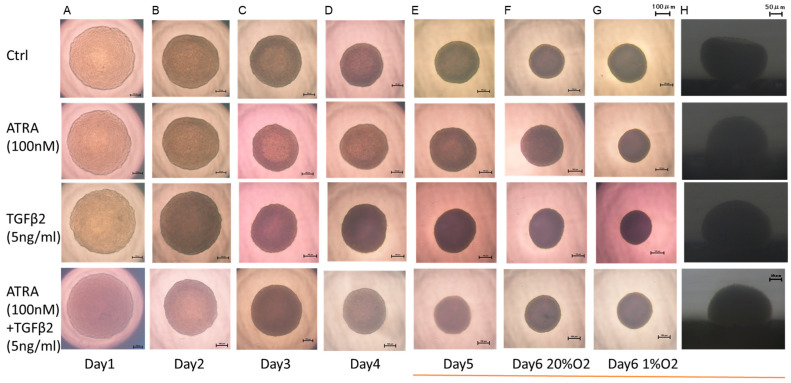
Effects of ATRA on 3D spheroid configuration of 3D ARPE19 spheroids not treated or treated with TGF-β2 under normoxia and hypoxia conditions. In the absence or presence of ATRA (100 nM), 3D ARPE19 spheroids not treated (Ctrl) or treated with TGF-β2 (5 ng/mL) were prepared under normoxia (20% O_2_) and hypoxia (1% O_2_) conditions. Representative phase contrast downward images ((**A**): day 1 under normoxia, (**B**): day 2 under normoxia, (**C**): day 3 under normoxia, (**D**): day 4, under normoxia) (**E**): day 5 under normoxia, (**F**): day 6 under normoxia, (**G**): day 6 under hypoxia) and lateral images ((**H**): day 6 under normoxia) are shown. Scale bar: 100 μm and 50 μm.

**Figure 6 biomedicines-12-02228-f006:**
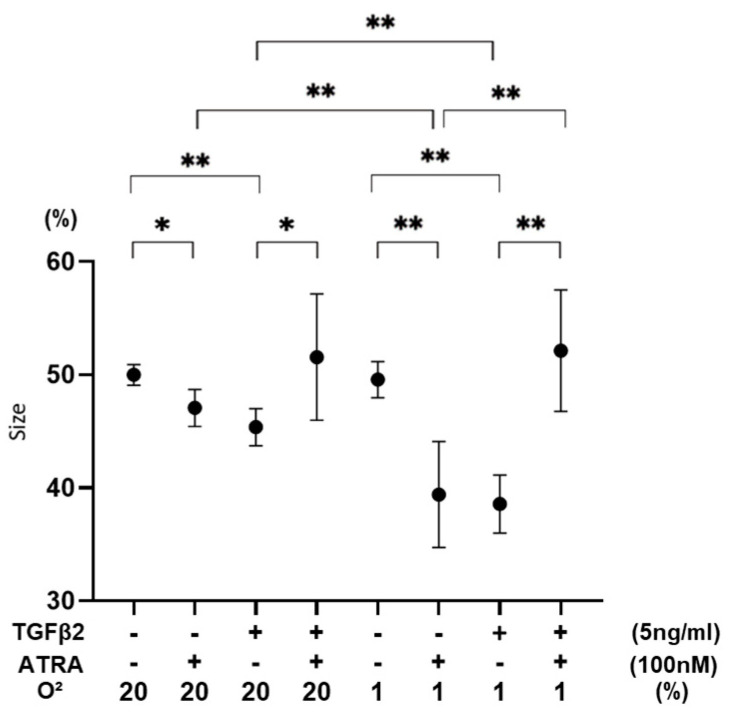
Effects of ATRA on 3D spheroid sizes of 3D ARPE19 spheroids not treated or treated with TGF-β2 under normoxia and hypoxia conditions. * *p* < 0.05, ** *p* < 0.01. In the absence or presence of ATRA (100 nM), 3D ARPE19 spheroids not treated or treated with TGF-β2 (5 ng/mL) were prepared under normoxia and hypoxia conditions. The mean sizes of 3D ARPE19 spheroids were measured and plotted. All experiments were performed in triplicate using fresh preparations consisting of 16 spheroids each. * *p* < 0.05, ** *p* < 0.01.

## Data Availability

The original contributions presented in this study are included in the article/Appendix A; further inquiries can be directed to the corresponding author.

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
