# Peer review of "Unexpected and Synergistical Effects of All-Trans Retinoic Acid and TGF-β2 on Biological Aspects of 2D and 3D Cultured ARPE19 Cells"

_biomedicines, 2024, doi:10.3390/biomedicines12102228_

Round 1

Reviewer 1 Report

Comments and Suggestions for Authors

The present paper provides interesting data on the complex functional interactions between ATRA (all-trans retinoic acid) and TGF-β2 on ARPE19 cells in physiological (normoxia) and pathological (hypoxia) condition.  The experimental protocol is clearly described and data obtained critically evaluated. It might be interesting to suggest functional hypothesis. Due to the relevant role plays by pigment epithelium in retinal function it might be interesting, in future, to have some experiment performed “in vivo” in control and degenerating retina.

Author Response

Dear Editor,

Thank you very much for the constructive comments concerning our manuscript “Unexpected and synergistical effects of all-trans retinoic acid and TGF-β2 on biological aspects of 2D and 3D cultured ARPE19 cells.”. We carefully checked all of the reviewer’s comments and prepared a revised version of our paper that takes these comments into account. The changes are listed below.

Reviewer 1

  1. The present paper provides interesting data on the complex functional interactions between ATRA (all-trans retinoic acid) and TGF-β2 on ARPE19 cells in physiological (normoxia) and pathological (hypoxia) condition. The experimental protocol is clearly described and data obtained critically evaluated. It might be interesting to suggest functional hypothesis. Due to the relevant role plays by pigment epithelium in retinal function it might be interesting, in future, to have some experiment performed “in vivo” in control and degenerating retina.

Answer; We sincerely appreciate your excellent comment. I totally agree with your excellent suggestion and therefore this information is included in the last paragraph of Discussion: ‘However, as of this writing, we do not know the reason for these unexpected synergistic effects of TGF-b2 and hypoxia, and therefore to better understand these unidentified issues, further investigations using 1) an RNA-sequencing analysis, 2) specific agonists and antagonists against down-stream factors of related signaling and 3) in vivo experiments using rodent models without or with degenerating retina will be required as our next project.’.

Reviewer 2 Report

Comments and Suggestions for Authors

Dear Authors,

Thank you for submitting your manuscript. I appreciate the effort you've put into this study. However, I have several suggestions that could enhance the clarity and overall quality of your paper:

Materials and Methods: Please add numbering to the subheadings in this section. This will improve the organization and make it easier for readers to follow.

Reagent Sources: It is essential to specify the sources of the reagents used, such as HG-DMEM and FBS. Including the supplier information adds credibility to your methods.

Results Section: Consider incorporating subheadings for each experimental result in the Results section. This will help readers quickly locate the data they are interested in and improve the flow of the section.

Figure Labeling: Please label the images in your figures using sequential identifiers (e.g., A, B, C, D). This is important for clarity, especially when referring to specific parts of the figures in the text.

Figure 2: The first two images in Figure 2 need to indicate statistical significance. Clearly marking significant results is critical for reader comprehension.

Summary Diagram: I recommend that you create a summary figure that encapsulates the experiments conducted in the study. A visual representation can greatly aid readers' understanding of your findings.

Discussion on Metabolic Changes: It is important to discuss whether there are shifts in cellular metabolic pathways between 2D and 3D ARPE cells under the same treatment conditions, as well as any corresponding changes in gene expression levels. This information would add depth to your discussion and provide valuable insights.

I hope these suggestions will help improve your manuscript. I look forward to seeing your revisions.

Best regards,

Reviewer

Comments on the Quality of English Language

Minor editing of English language required.

Author Response

Dear Editor,

Thank you very much for the constructive comments concerning our manuscript “Unexpected and synergistical effects of all-trans retinoic acid and TGF-β2 on biological aspects of 2D and 3D cultured ARPE19 cells.”. We carefully checked all of the reviewer’s comments and prepared a revised version of our paper that takes these comments into account. The changes are listed below.

Reviewer 2

Thank you for submitting your manuscript. I appreciate the effort you've put into this study. However, I have several suggestions that could enhance the clarity and overall quality of your paper:

  1. Materials and Methods: Please add numbering to the subheadings in this section. This will improve the organization and make it easier for readers to follow.

Answer; We sincerely appreciate your excellent comment. As suggested, numbering to the subheadings in this section is added.

  1. Reagent Sources: It is essential to specify the sources of the reagents used, such as HG-DMEM and FBS. Including the supplier information adds credibility to your methods.

Answer; We sincerely appreciate your excellent comment. As suggested, the supplier information of all chemicals is included.

  1. Results Section: Consider incorporating subheadings for each experimental result in the Results section. This will help readers quickly locate the data they are interested in and improve the flow of the section.

Answer; We sincerely appreciate your excellent comment. As suggested, subheadings for each experimental result in the Results section are included.

  1. Figure Labeling: Please label the images in your figures using sequential identifiers (e.g., A, B, C, D). This is important for clarity, especially when referring to specific parts of the figures in the text.

Answer; We sincerely appreciate your excellent comment. As suggested, sequential identifiers (e.g., A, B, C, D) are included in figures 2, 4 and 5.

  1. Figure 2: The first two images in Figure 2 need to indicate statistical significance. Clearly marking significant results is critical for reader comprehension.

Answer; We appreciate your careful review. In the mitochondrial respiration assay using the extracellular flux analyzer, statistical analyses should be performed only for the indices that were calculated using the representative measurements as listed Panels A and B in the revised Figure 2 in order to avoid multiple comparison problem if the metabolic indices are represented. We have clarified this point in the revised Figure legend. We sincerely hope that the reviewer will understand the standard statistic approach in the experiment of the mitochondrial respiration assay.

  1. Summary Diagram: I recommend that you create a summary figure that encapsulates the experiments conducted in the study. A visual representation can greatly aid readers' understanding of your findings.

Answer; We sincerely appreciate your excellent comment. As suggested, a summary figure demonstrating unexpected effects on physical properties of 3D ARPE19 spheroid by ATRA and TGF-b2 as a new Fig. 8.

  1. Discussion on Metabolic Changes: It is important to discuss whether there are shifts in cellular metabolic pathways between 2D and 3D ARPE cells under the same treatment conditions, as well as any corresponding changes in gene expression levels. This information would add depth to your discussion and provide valuable insights. I hope these suggestions will help improve your manuscript. I look forward to seeing your revisions.

Answer; We sincerely appreciate your excellent comment. As suggested, discussion related to metabolic changes in 3D cell cultures is included in last part of 1st paragraph of Discussion: ‘Comparison of cellular metabolism between 2D cells and 3D spheroids is intriguing. Although underlying mechanisms inducing such different effects of ATRA and/or TGF-β2 of ARPE19 cells and HTM cells between 2D and 3D culture conditions remain to be elucidated and that direct comparison between 2D cells and 3D spheroids is challenging be-cause of the presence of numerous different experiment conditions using the extracellular flux analyzer system, our previous studies showed that 3D spheroid culture induced spontaneous adipogenic differentiation of 3T3-L1 preadipocytes and a significant metabolic shift from glycolysis to mitochondrial respiration in differentiated 3T3-L1 cells, but those were not observed in the 2D cultured condition, suggesting that 3D cultured conditions served preferable environments for their adipogenic differentiation [50,51]. In addition, in our previous study using ARPE19 cells, we also reported that cellular metabolism in response to TGF-β2 was different between 2D cells and 3D spheroids [24] (Suzuki et al, PMID: 35628282). In that study, PGC-1α, which is one of master regulators of mitochondrial biogenesis, was significantly increased in 3D spheroids of ARPE19 compared to 2D cells of ARPE19, presumably partly via hypoxic stimulation. This suggests that mitochondrial OXPHOS may be more prominently activated in 3D spheroids under hypoxic conditions. However, it should be noted that the mechanism of metabolic changes in 3D spheroids are different depending on the types of cells [47,51,52,53] (Watanabe et al. PMID: 35740359, Endo et al. PMID: 37867843, Watanabe et al. PMID: 37511214, Nakamura et al. PMID: 38790973). Furthermore, an RNA sequencing analysis suggested that STAT3 signaling may induce these specific phenotypes observed in 3D cultured 3T3-L1 cells [50]. In fact, previous studies have shown that STAT3 signaling mechanisms are crosslinked with TGF-β2 [54], hypoxia related conditions [55] and ATRA [56,57]. Collectively, those synergistic effects of ATRA and TGF-β2 may support the well-accepted evidence that ATRA is involved in multifunctional and complicated effects in a variety of pathophysiological processes [11] despite the fact that metabolic adaptation in the 3D spheroids of ARPE19 cells has not been fully understood yet. Further study will be needed to elucidate the mechanisms why cellular metabolism in 3D spheroids show di-verse phenotypes.’

  1. Comments on the Quality of English Language: Minor editing of English language required.

Answer; We sincerely appreciate your excellent comment. Quality of English was carefully checked by a native English-speaking Scientist.

Round 2

Reviewer 2 Report

Comments and Suggestions for Authors

The authors have made thorough revisions in response to my comments, and I believe the manuscript has now met the publication requirements of the journal. However, the authors should note that there are a few citation errors in the newly added content of the Discussion section; it seems that the same reference is cited in two different formats. These need to be corrected before publication.

Author Response

Dear Editor,

Thank you very much for the constructive comments concerning our manuscript “Unexpected and synergistical effects of all-trans retinoic acid and TGF-β2 on biological aspects of 2D and 3D cultured ARPE19 cells.”. We carefully checked the reviewer’s comments and prepared a revised version of our paper that takes these comments into account. The changes are listed below.

Reviewer 2

The authors have made thorough revisions in response to my comments, and I believe the manuscript has now met the publication requirements of the journal. However, the authors should note that there are a few citation errors in the newly added content of the Discussion section; it seems that the same reference is cited in two different formats. These need to be corrected before publication.

Answer; We sincerely appreciate your valuable comment. As noted, there were citation errors in the newly added content of the Discussion section. We have carefully corrected them as suggested.